# Data-Driven Modeling of the Cellular Pharmacokinetics of Degradable Chitosan-Based Nanoparticles

**DOI:** 10.3390/nano11102606

**Published:** 2021-10-03

**Authors:** Huw D. Summers, Carla P. Gomes, Aida Varela-Moreira, Ana P. Spencer, Maria Gomez-Lazaro, Ana P. Pêgo, Paul Rees

**Affiliations:** 1Department of Biomedical Engineering, Swansea University, Swansea SA1 8QQ, UK; p.rees@swansea.ac.uk; 2i3S—Instituto de Investigação e Inovação em Saúde, Universidade do Porto, 4200-135 Porto, Portugal; carlag@ipatimup.pt (C.P.G.); info@i3S.up.pt (A.V.-M.); ana.spencer@i3s.up.pt (A.P.S.); maria.glazaro@ineb.up.pt (M.G.-L.); apego@ineb.up.pt (A.P.P.); 3Instituto de Engenharia Biomédica INEB, Universidade do Porto, 4200-135 Porto, Portugal; 4Faculdade de Engenharia da Universidade do Porto (FEUP), Universidade do Porto, 4200-465 Porto, Portugal; 5Instituto de Ciências Biomédicas Abel Salazar (ICBAS), Universidade do Porto, 4050-313 Porto, Portugal

**Keywords:** nanoparticle dosimetry, pharmacokinetics, imaging flow cytometry, nanomedicine, drug delivery, data-driven models

## Abstract

Nanoparticle drug delivery vehicles introduce multiple pharmacokinetic processes, with the delivery, accumulation, and stability of the therapeutic molecule influenced by nanoscale processes. Therefore, considering the complexity of the multiple interactions, the use of data-driven models has critical importance in understanding the interplay between controlling processes. We demonstrate data simulation techniques to reproduce the time-dependent dose of trimethyl chitosan nanoparticles in an ND7/23 neuronal cell line, used as an in vitro model of native peripheral sensory neurons. Derived analytical expressions of the mean dose per cell accurately capture the pharmacokinetics by including a declining delivery rate and an intracellular particle degradation process. Comparison with experiment indicates a supply time constant, τ = 2 h. and a degradation rate constant, b = 0.71 h^−1^. Modeling the dose heterogeneity uses simulated data distributions, with time dependence incorporated by transforming data-bin values. The simulations mimic the dynamic nature of cell-to-cell dose variation and explain the observed trend of increasing numbers of high-dose cells at early time points, followed by a shift in distribution peak to lower dose between 4 to 8 h and a static dose profile beyond 8 h.

## 1. Introduction

The development of nanoparticle-based drug delivery systems introduces a new pharmacological challenge. Whilst the incorporation of therapeutic molecules within a carrier matrix is well understood, the unique capability of nanoparticles (NPs) to enter cells means that the cellular dose of the drug carrier must be tracked and the subsequent delivery of the drug [1,2,3]. This is a complex task as drug delivery has many stages [4,5,6], in which particles cross multiple biological barriers [7,8] and release their cargo in response to cellular cues, environmental changes and/or (bio)degradation of the carrier [9,10]. Therefore, the description of nanoparticle pharmacokinetics requires new approaches that capture the dynamic nature of the multiple processes that control particle kinetics within cells [11].

The kinetics of drug release from a solid matrix have been exhaustively studied, providing a comprehensive understanding of this diffusion-driven process [12,13]. This knowledge has also been widely applied to nanoparticle matrices [14,15,16,17,18]. However, whilst the kinetics of a molecular cargo is well documented, there are relatively few reports on the modeling of nanoparticle pharmacokinetics/pharmacodynamics (PK/PD) at the cellular level. Most studies have focused on the organism level, with physiological-based pharmacokinetics (PBPK) in major organs [19,20] or tissue-based NP quantification [21,22]. Although models of the cellular accumulation of NPs and their link to cell death have been presented [23,24], there remains a need for flexible analytical models that accurately describe the processes of particle delivery, uptake, and drug release [25].

In this paper, we present mechanistic mathematical expressions and numerical simulations that describe the time dependence of nanoparticle dose at the cellular level. These accommodate various dynamic processes, controlling particle uptake and stability (degradability) within the cell, providing a theoretical model for interpreting nanoparticle pharmacological studies. The delivery system that we choose to model as an exemplar of our data-driven approach comprises trimethyl chitosan (TMC) NPs as gene delivery vectors designed for neuro-regenerative purposes applied to peripheral neuropathies. The simulated pharmacokinetics are validated by comparison to the previously reported data from studies using TMC NPs to vectorize plasmid DNA (pDNA). Chitosan (CH) is a biocompatible and biodegradable polymer and constitutes a promising gene delivery vector by its ability to complex and protect genetic material [26,27]. CH efficiency as a gene delivery vector relies on its molecular weight, which influences the formation of the NPs in terms of size and complexation capacity for oligonucleotide protection and transfection efficiency. CH solubility needs to be tuned for physiological conditions by chemical modification, and we have shown that CH quaternization constitutes a modification that results in increased solubility and transfection efficiency (e.g., trimethyl chitosan (TMC)) [27,28]. The constant fixed positive charges due to the quaternized amine groups allow nucleic acid complexation at physiological pH.

Furthermore, modifying the TMC degree of acetylation impacts the charge, solubility, binding capacity to nucleic acids, cellular uptake, and finally can result in an improved cellular breakdown of particles and controlled release of cargo. We found that the physicochemical characteristics of TMC–NPs show no significant differences as the degree of acetylation is changed. TMC was able to complex at least 90% of the pDNA, and TMC-pDNA NPs showed hydrodynamic sizes below 205 nm, low polydispersity index (PdI, <0.2), and positive surface charges (≥15 mV). To mimic in vitro conditions, TMC-based NPs were also evaluated after a 6-fold dilution in complete Dulbecco’s modified eagle medium (DMEM) at pH 7.4. Under these conditions, NPs’ features remained stable, presenting average sizes below 191 nm, PdI below 0.24, and positive surface charge (around +10.7 mV). After exploring the different degrees of acetylation, the best compromise between degradation kinetics and transfection efficiency was achieved with the TMC with a 16% degree of acetylation, which showed no cytotoxic effects [28].

The assessment of TMC-NP dose is based on cell images acquired with an imaging flow cytometer. The advantage of this high throughput imaging technique is that it allows the analysis of heterogeneous populations and enables access to every individual cell image. We present the data of Gomes et al. [28] in Figure 1, which shows the measured TMC-NP fluorescence (assumed to correlate to the cellular dose of pDNA cargo) at set time points over a 24-h period. Image panels from the imaging cytometer are included. From these, the evolution of dose as particles internalize to the neuronal cells can be seen. At the 0.5 h time point, the fluorescence appears as a diffuse signal across the whole of the cell as particles bind to the cell membrane’s outer surface; some early capture into cellular vesicles can be seen as punctate spots. After 4 h of exposure, the internalization process is well advanced, and most of the fluorescence signal comes from high-intensity spots corresponding to intracellular vesicles. The purpose of our data-driven analyses is to aid experimenters in the interpretation of time-dependent data such as those shown in Figure 1, for mean population measures and at the individual cell level. 

## 2. Materials and Methods

### 2.1. Cell Culture, Nanoparticle Preparation, and Image Acquisition

The details of cell culture, formulation of chitosan nanoparticles, exposure studies, and measurement of enzymatic degradation of the particles have been reported previously by Gomes et al. [28]. A full explanation of the materials and methods used can be found in this reference, and summary details are provided here.

TMC derived from ultra-pure CH was supplied by Kitozyme (40 kDa, Herstal, Belgium, lot VIHA0013-157). TMC was purified by filtration and dialysis. The pDNA was stained with the cell-impermeant dye YOYO^TM^-1 that binds and intercalates into the DNA (Invitrogen Corp., Waltham, MA, USA). YOYO^TM^-1 fluorescence increases over a thousand-fold when intercalated into double-stranded DNA, and it is used to image single DNA molecules. In this regard, the increased intensity of the YOYO^TM^-1 fluorescent image can be translated into a higher number of DNA molecules present and hence a higher number of NPs.

A sensorial neuron cell line model was used. The ND7/23 (mouse neuroblastoma (N18 tg 2) × rat dorsal root ganglion neuron hybrid) cell line was obtained from ECACC. The cells were routinely cultured in a complete DMEM medium (with serum and antibiotics) and maintained at 37 °C in a 5% CO_2_ humidified incubator. The ND7/23 cells were seeded into 24-well plates at a cell density of 2 × 10^4^ viable cells per cm^2^, 24 h prior to incubation with NPs. TMC-based NPs were prepared at an N/P ratio of 15 with 2 μg of pDNA per cm^2^ and incubated with cells from 0.5 h to 24 h, at 37 °C. TMC with 16% acetylation degree (TMC16) was used to prepare the TMC-based NPs. At each time point, cells were trypsinized and fixed with 4% paraformaldehyde (PFA). Cells were resuspended in PBS at a density of 10^5^ cells in 50 μL.

Cell images were acquired using an ImageStreamX multispectral imaging flow cytometer (Amnis, Luminex Corporation, Seattle, WA, USA.), collecting 10,000 cells per sample at 40 × magnification. A 488 nm wavelength laser was used to excite YOYO^®^-1, and the fluorescence images were collected using the 480–560 nm spectral detection channel.

### 2.2. Nanoparticle Characterization

Nanoparticle hydrodynamic mean diameter and zeta potential were determined using a Zetasizer Nano Zs (Malvern Instruments, Malvern, UK) following the manufacturer’s instructions. NPs were diluted 6 times in Dulbecco’s Modified Eagle Medium (DMEM) with Glutamax™, 10% (*v*/*v*) heat-inactivated (55 °C) fetal bovine serum (FBS) and 1% (*v*/*v*) penicillin/streptomycin, all supplied by Gibco (Thermo Fisher Scientific, Waltham, MA, USA), and left to stabilize for 30 min prior to NP size and surface charge determination, using a high concentration cell (ZEN1010, Malvern Instruments, Malvern, UK). Size measurements were performed at 25 °C at a 173° scattering angle in the automatic mode, and the mean hydrodynamic diameters were determined by cumulant analysis (Z-average mean). The calculated zeta potential of the NPs was determined by laser Doppler velocimetry (LDV) and phase analysis light scattering (M3-PALS) at 25 °C. The software used was DTS Nano version 7.11, supplied by the manufacturer (Malvern Instruments, Malvern, UK)

### 2.3. Nanoparticle Supply Kinetics

To assess the rate of delivery of nanoparticles to cultured cells, the nanoparticle cargo (DNA) was labeled with the fluorescent probe, YOYO^®^-1 (Invitrogen Corp., Waltham, MA, USA) (as described previously [28]) and applied in wells at a 1.9 nM particle concentration. The fluorescence signal generated by particles on the bottom of the well was measured at hourly intervals over a 16-h period, using a confocal microscope (integrated intensity of the bottom of the well with a pinhole of 6 a.u.). The integrated intensity of the bottom of the well was calculated at each time point using ImageJ software (version 1.52, National Institutes for Health, Bethesda, MD, USA).

### 2.4. Analytical Models

A rate equation analysis was used to derive expressions for the time-dependent signal per cell arising from labeled nanoparticles (signal assumed to be proportional to particle number). Different cases are presented, dependent upon the specific conditions assumed for particle supply and loss processes.

A1. Simple case of nanoparticle accumulation from an infinite supply reservoir:(1)$$\frac{dS}{dt}=a$$
where *S* is the measured dose signal (arbitrary units), and *a* is the accumulation rate (per second).
(2)$$\int_{0}^{S}dS=\int_{0}^{t}a\;dt$$
(3)S=at

There is a linear increase in signal at a rate, *a*.

A2. Nanoparticle accumulation from an infinite supply but with a dose loss mechanism:(4)$$\frac{dS}{dt}=a-bS$$
where *b* is the rate of signal loss per unit metric.
(5)$$\int_{0}^{S}\frac{1}{a-bS}dS=\int_{0}^{t}dt$$
(6)$$S=\frac{a}{b}\left(1-e^{-bt}\right)$$

The (1 − exp) term produces a non-linear signal increase to a saturation level equal to *a*/*b*. This maximum reflects a steady state in which the number of particles accumulated per unit time equals the number lost.

A3. Nanoparticle accumulation in the presence of a supply limitation:(7)dSdt=ae−tτ
where *τ* is the decay time-constant of an exponentially decreasing supply rate.
(8)∫0SdS=a∫0te−tτdt
(9)S=[−aτe−tτ]0t
(10)S=aτ(1−e−tτ)

The signal saturates at a level a/τ, as the particle reservoir becomes depleted and supply ceases.

A4. Nanoparticle accumulation in the presence of supply limitation and particle loss:(11)dSdt=ae−tτ−bS
(12)dSdt+bS=ae−tτ

Using an integrating factor: *e^bt^*
(13)∫ ebtdSdt+∫ ebtbS=∫ aebte−tτ
(14)Sebt=−a(1τ−b)e−t(1τ−b)+K

At *t* = 0, *S* = 0 therefore, K=a(1τ−b) and:(15)S=a(1τ−b)[1−e−t(1τ−b)]ebt

A complex function with countervailing terms describing signal growth/decay driven by particle accumulation/loss.

## 3. Results

### 3.1. Population-Averaged Pharmacokinetics

Our starting point for the analysis was estimating the mean level of particle accumulation across a cell population. The complexities of biological heterogeneity were avoided by treating cells as a homogenous group to derive a population average dose. We aimed to identify the key generic processes of dose accumulation and intra-cellular transformation. The focus is on process kinetics. It is the temporal evolution of dose that we wish to describe. Thus, the analysis described the outcome rather than the specific biological mechanism at play (e.g., endocytosis, diffusion, etc.). Even under the constraint of a population mean, the kinetics of particle supply and trans-membrane transport, together with the instability of degradable nanoparticles, present a multiplicity of potential system dynamics. The analytical expressions presented in the methods section describe three possible scenarios:

1.Particle uptake at a constant rate (A1)

The simple situation of stable particles and a constant internalization rate for cellular uptake.

2.Particle uptake with a limiting process to supply or accumulation (A2 and A3)

This introduces the possibility of bottlenecks or loss mechanisms that operate to limit the nanoparticle dose. Two possible situations are considered, a rate reduction due to limitations in particle supply or a loss process that reduces the cellular dose post uptake (e.g., nanoparticle disassembly or exocytosis).

3.Particle uptake with a limiting process to supply and accumulation (A4)

The case for accumulation of particles in the presence of a bottleneck in supply and a cellular loss in dose was considered.

The time-dependent accumulation predicted by each of these models is shown in Figure 2. In the ideal case of an unlimited supply of NPs and a constant internalization rate, there is a linear increase in particle number (fluorescence signal) over time. This is seldom achieved as mechanisms that limit the maximum dose result in a mean number of particles per cell that rises and then saturates to a fixed level. The time-dependent dose curve follows a [1 − exp(−κt)] form, and this kinetic has been observed for various limiting processes. In NP assays, supply is often limited as a fixed environmental reservoir of particles is significantly depleted by transfer to cells [29,30]. This form of saturation in delivered dose is also seen for drug delivery from a solid matrix [15,31], where similar limitations due to a depleting supply take effect. When the dose limitation is due to loss from the cell, the saturation level reflects the situation where rates of particle internalization and loss are equal. This type of loss process will arise when there is intra-cellular loss, e.g., in the case of biodegradable particles [32], or can result from particle expulsion, e.g., such as occurs when partitioning into daughter cells upon cell division [23] or exocytosis [33,34]. Finally, the loss mechanism could relate to a lack of internalization rather than loss of particles per se. The kinetics of binding and release of ligands with cell surface receptors can follow a [1 − exp(−κt)] form as available binding sites become saturated [35,36,37]. In the presence of supply limitations and particle loss, three processes drive the system dynamic—arrival from a depleting supply reservoir, internalization, loss from the cell, and hence we see an additional inflection in the accumulation curve. The cellular loss of a finite supply of particles drives the dose to zero over long periods.

These fundamental processes shaping the particle dose curve provided preliminary analysis for an NP dosing study. It highlighted limitations to the long-term cellular accumulation of particles and indicated the process time-constants involved. As an example, the data from the study of Gomes et al. [28] is shown in Figure 3 (fluorescently labeled TMC16 particles—16% acetylation, incubated with ND7/23, neuronal cell line). The time-dependent fluorescence of these trimethyl chitosan particles exhibited multiple dynamics with initial accumulation followed by reduction to a stable level within 24 h (Figure 3A). The study used biodegradable nanoparticles, specifically designed to dissociate when in cellular vesicles. There was, therefore, an imposed particle loss mechanism. This process alone would lead to a monotonically increasing dose that saturates over time (Figure 2 and Materials and Methods, A2, Equation (6)). It is clear, therefore, that other processes were at play here. The kinetics of the particle degradation were captured using a competition assay in which the particles were incubated with lysozyme and a fluorogenic substrate [28,38] (Figure 3B). The action of the enzyme led to the breakdown of the chitosan particles resulting in a decrease in the fluorescent signal. Notably, there was not a complete loss of the signal, and we attribute the fixed baseline to a stopping point, beyond which particles do not undergo further enzymatic degradation. This limitation to the action of the lysozyme has been attributed to the loss of acetylated units during degradation leading to a lack of binding sites for the enzyme once the chitosan reaches a baseline molecular weight [28]. Fitting the time-dependent fluorescence data using an exponential decay model (Materials and Methods, A2) indicated a rate-constant, *b* for the particle degradation of 0.71 per hour and a fixed, time-independent signal equal to 0.38 of the maximum value. To assess the supply kinetics TMC-NP with a fluorescently labeled DNA cargo were introduced to the cell culture wells. The fluorescence signal from the bottom of the well was then measured in the absence of cells (Figure 3C). This provided a measure of the supply rate, independent of cell internalization processes. The signal’s unambiguous saturation, indicating complete cessation of particle supply from the suspension, with a rise-time constant, *τ* of 2 h (Materials and Methods, A3, Equation (10)).

The combined action of this supply bottleneck and the cellular degradation of nanoparticles is sufficient to explain the time-dependent dose profile. Substitution of the measured rate constants for these processes into the analytical functions provides an accurate representation of the mean fluorescence per cell (Materials and Methods—A3, Equation (10) used for the stable fluorescence signal and A4, Equation (15) for the biodegradable fraction, in a 38:62 ratio). There is a single fit-variable of the accumulation rate, a, which scales the model curve relative to the dataset (Figure 3A). The strong correlation between the analytical prediction and the measurement data confirms the model’s validity as a predictor of particle dose kinetics. It supports the hypotheses implied in forming the analytical equations.

### 3.2. Single Cell Pharmacokinetics

Biological heterogeneity ensures that there will be a variation of acquired dose at a cellular level for a single NP exposure concentration. Thus, while prediction of the mean dose per cell is informative, a detailed understanding of particle accumulation requires analysis and quantification of the cell-to-cell dose variability. The predominant experimental technique for studying the cellular dose of particulates is flow cytometry which naturally captures and visualizes measurement variance using histograms. This suggests similar approaches to the modeling of single-cell pharmacokinetics, with a focus on data distributions. This would allow direct interpretation of simulations and an intuitive understanding of their import as they are presented in the same format as the experimental data. The data-directed approach also provided a novel route to model construction as the time-dependent evolution of the data construct could be simulated rather than the data itself, i.e., the changing shape of the histogram is predicted, not the values of the measured variable—individual cellular dose. It is this direct simulation of the cytometry histogram that we adopted. The principles of our approach are demonstrated in Figure 4, which depicts the algorithmic transformations for a selected subset of bins.

An initial (t = t_0_) distribution of dose per cell was created using a probability distribution function (PDF) to reflect the heterogeneity of uptake per unit timestep, Δt [39] (Figure 4A). The exact fitting of experimental data requires modeling variance in cell area and particle arrival rate, resulting in an asymmetrical, over-dispersed PDF [40]. Here we took a simplified approach and adopted a Poisson distribution function, with the rate variable, λ, representing the mean dose accumulated over Δt. Although this did not match measured cytometry histograms exactly, it provided the important accounting of variability in particle uptake. It allowed efficient and plausible simulation of the time-dependent evolution of histogram shape and the overall mean value of dose. 

The cell dose’s time dependence is modeled by manipulating the histogram bins to simulate the effect of continued dose accumulation and the ensuing biodegradation within cell vesicles. Over a timestep (t_0_ → t_0_ + Δt), cells in bin *i* with dose *D_i_* will continue to accumulate particles. However, due to the heterogeneity of uptake, there will be a dispersion in the dose as this sub-set of cells, with identical dose at t_0_, gain differing numbers of particles. This is simulated by replacing the single bin-value of *N_i_* cells of dose *D_i_*, with the Poisson dose-accumulation PDF, offset to a mean of (*D_i_* + λ) and with a normalized area of *N_i_* to conserve cell number (Figure 4B). The biodegradation of nanoparticles leads to a fractional loss, *η* in dose over the period (*η* = *e*^−bΔt^), which is accounted for by a leftwards shift of the histogram with new bin values of *η*.*i* units (Figure 4C). The total dose distribution is calculated from implementing the above procedure for each bin (*i =* 1:n), followed by the summation of the n histograms produced (Figure 4D). To simulate the decrease in particle uptake rate stemming from depletion of the supply reservoir, the Poisson distribution rate is implemented as a time-dependent parameter, λ(t), exponentially decreasing according to the measured time constant, *τ* (see Figure 3C).

The net result of these dose-determining processes is increasing or decreasing cell number counts across the histogram bins over time. The signature of the pharmacokinetics can thus be seen in the changing shape of the distribution. A comparison of the data-histogram simulation to experimental distributions is shown in Figure 5 for cells incubated with TMC16 particles over a 24-h period. 

The parameter values for the time constant, *τ* of depleting particle supply and rate constant, *b* of particle degradation, were obtained from the population-averaged dose measurements, i.e., they are experimentally determined quantities (see Figure 3). The simulated histogram is produced using the mathematical transformations described above and includes a fixed component to describe the stable fluorescent signal that remains unchanged over time (38% of the total). The model data set provides an accurate simulation of the experiment, with the evolution of distribution shape seen to be the same across the sets. The benefit of the model is that the detail of the evolution in dose variance can be interpreted with the knowledge of the physical processes on which it is based. Over the first few hours of the experiment, the dose per cell increases, as would be expected, and the histogram shifts rightwards along the *x*-axis. However, by the 8-h time point, the combined effect of particle degradation and a diminishing supply from the medium leads to a lowered dose per cell. This manifests in the distribution as reduced cell numbers in the high dose bins and increasing numbers in the low dose bins. This shifts the peak and skews the shape of the distribution leftwards along the *x*-axis. While particle arrival at the cell does occur beyond the 8-h time point, it is minimal (see Figure 3C), and the fluorescence of previously acquired particles has decayed to the baseline value by this time (see Figure 3B). The influence of particle biodegradation and supply on the overall dose kinetic is complete by the 8-h mark. So there is little further change in the histogram beyond this point as it represents a stable dose within cells that are no longer exposed to particles from the surrounding medium. This agrees with previous measurements of chitosan internalization in this cell line [28].

## 4. Discussion

The development of active nanosystems, designed to deliver targeted interaction with selected cell types and exploit intra-cellular processes to release a drug cargo, greatly adds to the complexity of their pharmacokinetics. This necessitates a reappraisal of standard assumptions on the scalability of exposure and dose, encapsulated in commonly applied rules such as Haber’s law [29,41]. The multiple stages and biological interactions of nano-pharmacology led to a dose-time relationship markedly non-linear and potentially non-monotonic. These functional complexities may be viewed as confounding factors to the understanding of pharmacology. However, the very fact that the various particle-cell interactions shape the dose profile means that their signature may be extracted from the data. In this paper, we present analytical functions suitable to describing the mean dose-time curve for cell populations. The set of equations provided is an easily adopted tool to identify key processes at play and provide a first approximation of dose kinetics in an experiment from standard measures of mean cell dose. The equations describe supply variation and loss of intra-cellular dose in cells exposed to multi-stage NP drug-delivery systems. The model parameters: *a* (dose accumulation-rate) and *b* (intra-cellular dose loss-rate) describe generic processes common to most NP exposure conditions. They are agnostic to the specific mechanisms at play, i.e., we do to attempt to describe how NPs enter the cell or how they may be transformed by cellular metabolism. We aimed to identify the signature of kinetic processes within time-dependent data and provide the experimenter with tools to aid the understanding of these patterns.

The validity of these analytical models is demonstrated by their application to a study of the uptake of biodegradable chitosan particles by neuronal cells. This specific example highlights limitations to the drug supply due to the depletion of NP concentration in the cellular environment and the loss of particles once internalized. The correspondence of the mathematical prediction to the data (see Figure 3A) suggests that in this case, the quantified degradation of particles by cell vesicle enzymes is sufficient to account for the reduction in mean dose with no apparent contribution from particle exocytosis.

The interpretation of cellular pharmacokinetics is usually based on dose histograms from flow cytometry. From the point of view of applicability, the ideal model should simulate this primary experimental data and mimic the cell-to-cell heterogeneity inherent in measured dose distributions. We adopt this premise and bring a fresh approach to cell modeling by directly simulating the data construct rather than the measurement variable itself—the time-dependent rise and fall in cell populations of a specified dose is modeled by direct manipulation of bins in a simulated histogram. Modeling the data plot in this way guarantees relevance to the experiment and facilitates interpretation of the simulation results. It also restricts the computational task to implementing mathematical transformations on a set of distribution bins. This is simpler than the commonly used techniques. A multiplicity of rate equations is used to describe heterogeneous cell sub-groups, or a stochastic, ‘biological-noise’ term is included within a single global rate equation to account for process variability [42,43,44].

As nanomedicine has developed, there has been a continual advancement in metrology techniques for assessing NP dose, often in response to identified gaps in current capability [45,46]. The evolution of multi-functional, smart nanoscale drug delivery systems that interact with endogenous biological processes presents a challenge to measurement techniques. The important additionality here is that for these highly dynamic systems, the pharmacokinetic process must be quantified and pharmacological dose. The methods presented in this paper provide easily adopted approaches to this challenge, developed as data-driven tools to ensure ready interpretation of experimental results.

## Figures and Tables

**Figure 1 nanomaterials-11-02606-f001:**
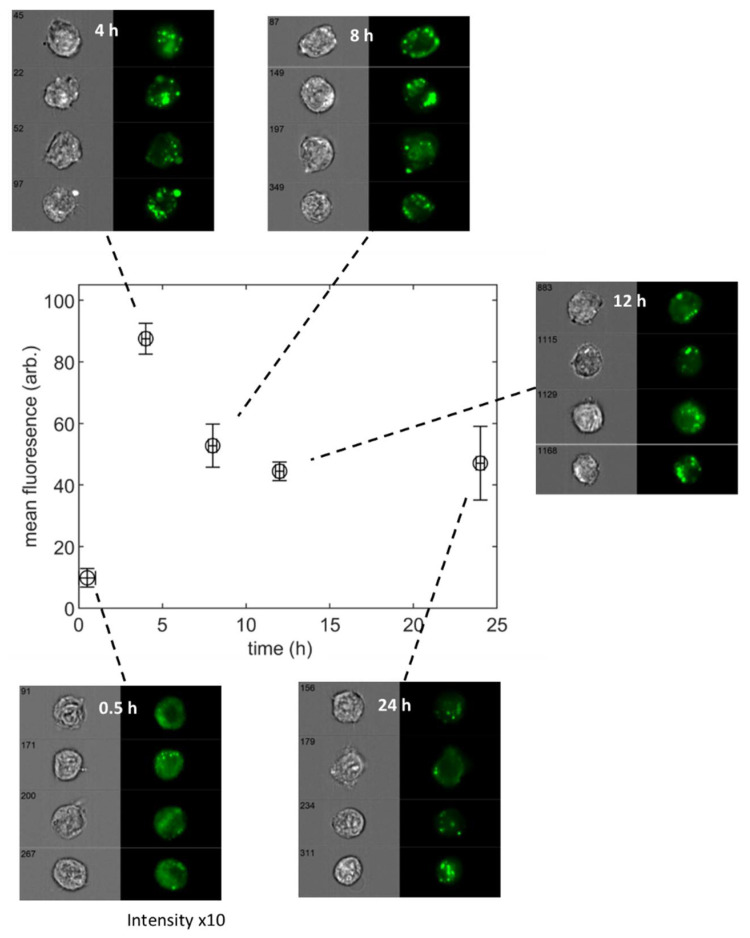
Mean YOYO^TM^-1 fluorescence (p-DNA marker) per cell from dosing study with ND7/23, neuronal cells exposed to trimethyl chitosan (TMC16) nanoparticles (data from Gomes et al. [28]). The linked panels contain representative bright field and fluorescence images of the cells at each measurement time point (image numbers refer to cell reference in the imaging acquisition stream). The pixel intensity of the fluorescence channel is scaled by a factor of 10 for the 0.5 h time point to enhance visibility. All other images have the same pixel intensity range.

**Figure 2 nanomaterials-11-02606-f002:**
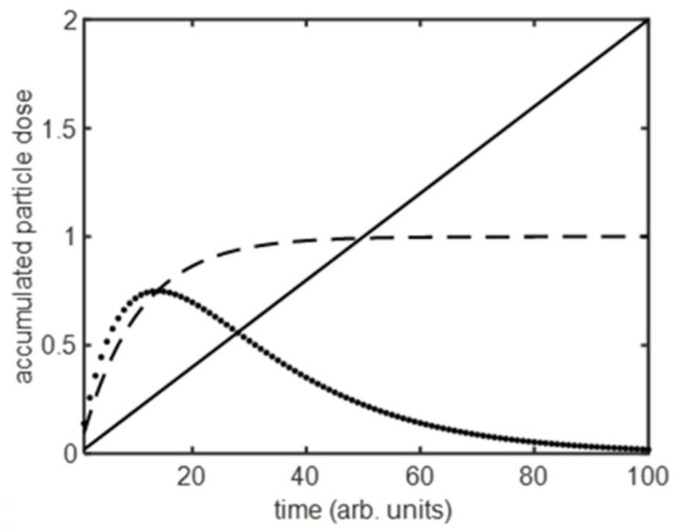
Schematic of the time-dependent cellular dose of nanoparticles under different conditions; i. accumulation at a constant rate from an infinite particle reservoir (solid line); ii. accumulation with a supply rate limitation or particle loss from the cell (dashed line); iii. accumulation with supply rate limitation and particle loss from the cell (dotted line).

**Figure 3 nanomaterials-11-02606-f003:**
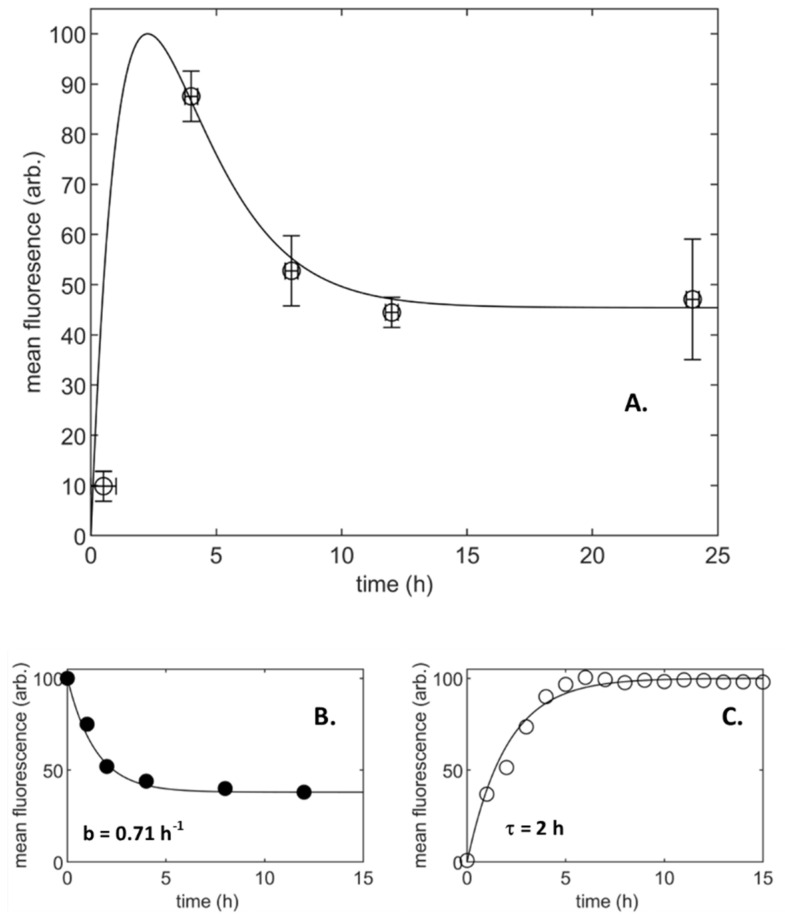
(**A**). Mean fluorescence per cell from dosing study with ND7/23, neuronal cells exposed to trimethyl chitosan (TMC16) nanoparticles (data from Gomes et al. [28], solid line: fit curve using the analytical function in Methods—A4, error bars indicate s.e.m.). (**B**). fluorescence from nanoparticles undergoing enzymatic degradation (data from Gomes et al. [28], solid line: *e*^−*bt*^ fit curve). (**C**). accumulation of particle fluorescence signal from the base of the culture well (solid line: [1 − *e*^−*t/τ*^] fit curve).

**Figure 4 nanomaterials-11-02606-f004:**
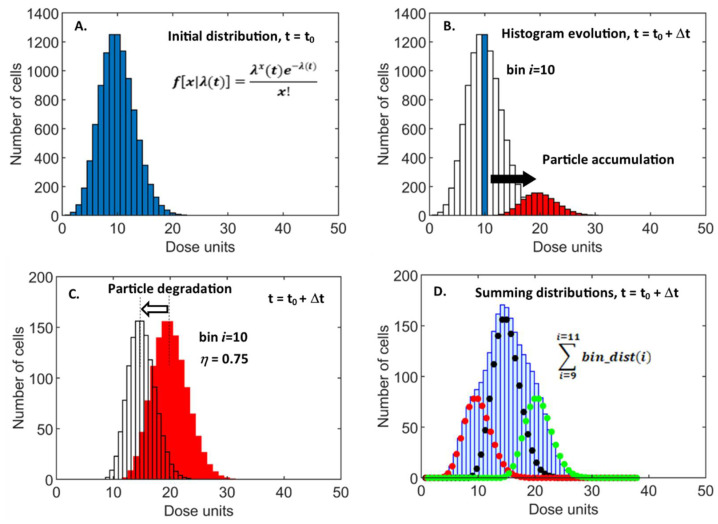
Schematic of histogram evolution to simulate particle accumulation and biodegradation. (**A**). Initial Poisson PDF generated at t_0_. (**B**). Continued NP accumulation within cells of bin 10 during t to t + Δt shown. (**C**). Reduction in dose by a fraction η, due to NP degradation. (**D**). final summing of dose at t + Δt. The example is shown for a limited sample of 3 distributions, corresponding to original bin indices 9, 10, and 11, at t_0_.

**Figure 5 nanomaterials-11-02606-f005:**
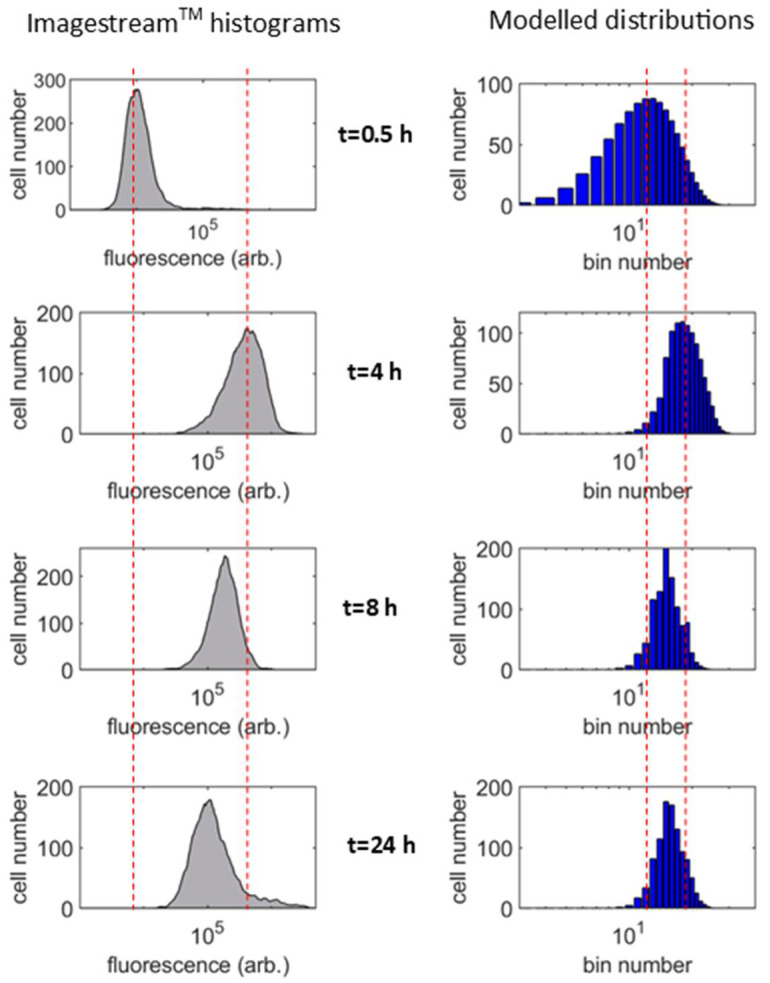
Measured histograms of fluorescence intensity per cell [28] (left column) and simulated distributions (right column), for ND7/23 neuronal cells incubated with TMC16 nanoparticles. Red lines provide visual reference points, against which histogram movement can be assessed.

## Data Availability

The data presented in this study are available in the previously pblished paper: Acta Biomater. 2016, 46, 129–140, https://doi.org/10.1016/j.actbio.2016.09.037.

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
