# Peer review of "Data-Driven Modeling of the Cellular Pharmacokinetics of Degradable Chitosan-Based Nanoparticles"

_nanomaterials, 2021, doi:10.3390/nano11102606_

Round 1

Reviewer 1 Report

In the proposed paper, the authors presented a mechanistic mathematical expressions and numerical simulations describing the time dependence uptake of trimethyl chitosan (TMC) nanoparticles, labelled with pDNA within a sensorial neuron cell line model (ND7/23). In particular, in order to evaluate the uptake of  TMC nanoparticles in the cultured cells, the NPs was further labelled with a fluorescent probe (YOYO®-1) and the time dependence uptake was measured by means of  flow cytometer. The authors proposed a set of equations describing possible uptake mechanisms of nanoparticles in cells. The mathematical predictions fit with the experimental findings reported in the paper, confirming the validity of the model as a predictor of particle dose kinetics.

The manuscript might be interesting and relevant to the drug delivery and nanomedicine community, however, it requires changes before publication. There are some relevant formal and technical issues that must be addressed.

  1. The article is well written and no grammatical errors are found but it is currently challenging to read. The authors should revise the manuscript providing a better description of the research background/state of art and adding more information on fundamental aspects of the proposed research paper, such as particle size/shape, uptake mechanism, localization of nanoparticles in the cell and intra-cellular degradation process. I am aware that these information are contained in their previous article http://dx.doi.org/10.1016/j.actbio.2016.09.037 ) but, in my opinion, It is necessary to briefly report these information in this manuscript to facilitate the reading and understanding of the article.
  2. The authors should clarify why they used in this work the mouse sensorial neuron cell line model ND7/23. Is there a specific reason?
  3. At page 4, the authors state that “By treating cells as a homogenous group to derive a population average dose, the complexities of biological heterogeneity are avoided, with the hope that despite this simplification the fundamental processes driving particle-cell interaction can still be elucidated.” In my opinion, that description is not accurate and the authors should avoid statements such “with the hope that” in a scientific paper. Moreover, the authors should better clarify and report the nanoparticles uptake mechanism.
  4. The authors did not report in the manuscript any image (acquired with an optical, fluorescence or confocal microscopy) of the cells treated at different incubation times. These images could support the conclusions and results presented. Furthermore, the images of the cells could provide fundamental information regarding the state of the cells after treatment (for example: alterations in cell morphology induced by NPs treatment).
  5. The authors should briefly indicate the techniques for the morphological characterization of nanoparticles.
  6. The reference [4] is incomplete.

Author Response

Please see our response in the attached word document

Reviewer 2 Report

Dear Editor, in the submitted paper authors presented a validation study which simulates the time-dependent dose of trimethyl chitosan nanoparticles in an ND7/23, neuronal cell line. The paper is oversimplified with limited studies and available data. I think that additional experiments should be done in order to be characterized as a completed study. In following you can find my specific comments.

  • Some characteristics of chitosan nanoparticles can be also presented here.
  • Why in this study only trimethyl chitosan nanoparticles were used?
  • Which are their advantages compared with neat chitosan nanoparticles or compared with other derivatives? This is not clear and lots of chitosan derivatives have been prepared and mentioned in literature.
  • How is particle size of the used nanoparticles affected to interactions with selected cell types? This is very crucial since changing the particle sizes the most of nanoparticles characteristics are changing.
  • It is clear that limited parameters have been used to authors chosen models.

Author Response

REEREE 2

Dear Editor, in the submitted paper authors presented a validation study which simulates the time-dependent dose of trimethyl chitosan nanoparticles in an ND7/23, neuronal cell line. The paper is oversimplified with limited studies and available data. I think that additional experiments should be done in order to be characterized as a completed study. In following you can find my specific comments.

Some characteristics of chitosan nanoparticles can be also presented here.

Why in this study only trimethyl chitosan nanoparticles were used?

As stated in the manuscript, the current paper explores data previously published to propose a new mathematical model that can predict the cellular pharmacokinetics of degradable chitosan-based (TMC) nanoparticles (see Gomes, C.P.; Varela-Moreira, A.; Leiro, V.; Lopes, C.D.F.; Moreno, P.M.D.; Gomez-Lazaro, M.; Pêgo, A.P. A high-throughput bioimaging study to assess the impact of chitosan-based nanoparticle degradation on DNA delivery performance. Acta Biomater. 2016, 46, 129–140, doi:10.1016/j.actbio.2016.09.037.). In our previous study we explored the quaternization of chitosan to improve the performance of chitosan as a nucleic acid delivery system. As previously reported, quaternization not only improves the solubility of chitosan at physiological pH but also the stability of DNA-chitosan complexes. Consequently, this has been the selected type of chitosan derivative to be explored in future studies. In the work presented in this paper our purpose is not to compare the capacity of different particles as drug delivery carriers but rather to show a new methodology to interpret the time dependent behaviour of these NPs in their interaction with cells. Nevertheless, to aid the reader we have added further information on the advantages of TMC over chitosan was included in the Introduction section.

Which are their advantages compared with neat chitosan nanoparticles or compared with other derivatives? This is not clear and lots of chitosan derivatives have been prepared and mentioned in literature.

As stated above, the aim of this study was not to conduct a comparative study of the performance of different chitosan based materials. Nevertheless, we have enhanced the paper with detailed information of the nanoparticles used as well as the rationale for the selection of TMC as delivery system (see Introduction).

How is particle size of the used nanoparticles affected to interactions with selected cell types? This is very crucial since changing the particle sizes the most of nanoparticles characteristics are changing.

We agree with the reviewer that it is known for chitosan-based nanoparticles with pDNA cargo that the particle size and charge are important for the pharmacokinetic and biodistribution profiles. Also, cellular uptake kinetics can be modified by the quaternization of chitosan (as compared with similar NPs sizes). We have added text around these points to the introduction.

Our paper presents computational and mathematical methodologies for assessing nanoparticle dose dynamics, the analytical approaches are the primary focus of the manuscript. We have demonstrated application of our techniques by reference to previously reported experimental studies on chitosan nanoparticle uptake. Comprehensive discussion on the aspects raised by the referee can therefore be found in the source article by Gomes et al, reference 28 in our manuscript - 

Gomes, C.P.; Varela-Moreira, A.; Leiro, V.; Lopes, C.D.F.; Moreno, P.M.D.; Gomez-Lazaro, M.; Pêgo, A.P. A high-throughput bioimaging study to assess the impact of chitosan-based nanoparticle degradation on DNA delivery performance. Acta Biomater. 2016, 46, 129–140, doi:10.1016/j.actbio.2016.09.037.

We recognise that reference back to another document disrupts the reading process and so we have added additional text in our introduction to provide some context on the use of chitosan particles.

It is clear that limited parameters have been used to authors chosen models.

Our purpose is to present a general analytical approach through which an understanding of process in nanoparticle uptake experiments can be gained. The mathematical and computational techniques are what we are aiming to present as the core focus of the paper. The models are general as they are based on ubiquitous processes of dose accumulation and intra-cellular transformation, they are therefore applicable to a wide range of experimental conditions. We use this particular experimental data set to provide a specific example of implementation for the reader. We also emphasise that all model parameters come directly from experimental measurement and so there is no selection to a pre-determined model configuration.

Reviewer 3 Report

Overall, the work described in this manuscript is very interesting lying on the fact that very little information is available to understand the fate of nanoparticle at the cellular level. However, in the current state in which the paper is written is pretty difficult to follow which data has been taken into account to perform the pharmacokinetic profile. Also, it would be interesting if you reproduce partially the data from the other paper you are referring to (asking for the corresponding permissions) so actually, the readers can fully understand what you are calculating and what you are trying to predict. If I read well, are the pharmacokinetic experiments based on confocal measurements? How accurate can be these measurements? Don't you think that HPLC or LC-MS could be a better analytical technique to quantify the amount of drug internalised by the cells? Also, how can you differentiate between drug adsorbed in the surface of the cells and drug fully internalised? How have you quantified drug degradation? Have you used a stability indicating HPLC method?

Also in the mathematical equations, you should take into account transformation or metabolism of the drug as it can trigger higher activity or toxicity. Limitations of this methodology should be well described in the discussion and potential solutions or further considerations that should be taken into. 

Author Response

(The authors gave the same response as above.)

Round 2

Reviewer 1 Report

Dear Authors,

Thanks for your answers. I believe that the answers you have provided and the modifications to the original manuscript are comprehensive, exhaustive and facilitate the reading and understanding of the article. Personally, I greatly appreciated the changes made in the introduction of the articulations and the addition of Figure 1. 

I believe that you have fully answered my doubts and requests for clarification

Best Regards,

Paolo

Author Response

We thank the reviewer for their feedback on the manuscript.

Reviewer 2 Report

Dear Editor, I have seen the revised manuscript and authors response. It seems that everything is fine with their work and nothing new should be added from the proposed experiments. The papers deal with data from another worked published many years ago!!! Why there is a need to do this study after 5 years???? There are some texts added to the introduction of this work but nothing to important in order to change the paper and to enhance its scientific value.  My opinion is again that this paper is of low importance and additional experiments should be done in order to be state as a complete work.

Author Response

We thank the reviewer for their feedback on the manuscript. The reason for this work is to introduce new mathematical and computational techniques. It is this modelling that is the key message of the paper.